# Systemic Antifungal Therapy for Invasive Pulmonary Infections

**DOI:** 10.3390/jof9020144

**Published:** 2023-01-21

**Authors:** Ronen Ben-Ami

**Affiliations:** Infectious Diseases Unit, Tel Aviv Sourasky Medical Center, Faculty of Medicine, Tel Aviv University, Tel Aviv 6997801, Israel; ronenba@tlvmc.gov.il; Tel.: +972-3-6974347; Fax: +972-3-6974996

**Keywords:** antifungal drug, antifungal resistance, invasive aspergillosis, mucormycosis

## Abstract

Antifungal therapy for pulmonary fungal diseases is in a state of flux. Amphotericin B, the time-honored standard of care for many years, has been replaced by agents demonstrating superior efficacy and safety, including extended-spectrum triazoles and liposomal amphotericin B. Voriconazole, which became the treatment of choice for most pulmonary mold diseases, has been compared with posaconazole and itraconazole, both of which have shown clinical efficacy similar to that of voriconazole, with fewer adverse events. With the worldwide expansion of azole-resistant *Aspergillus fumigatus* and infections with intrinsically resistant non-*Aspergillus* molds, the need for newer antifungals with novel mechanisms of action becomes ever more pressing.

## 1. Introduction

Human invasive fungal diseases (IFDs) have increased in frequency and diversity over the past three decades [1,2], driven by an ever-expanding population of immune-compromised patients and the increasingly appreciated effects of climate change [3,4]. The respiratory system is the principal point of entry for airborne fungal spores and, consequently, the primary site of most IFDs. Most systemic antifungals used in the treatment of IFD belong to only three drug classes: polyenes, azoles and echinocandins. Amphotericin B, the only polyene available for systemic use, was introduced in 1958, and represented the main treatment option for pulmonary mold diseases for more than four decades [5]. The treatment of invasive aspergillosis was revolutionized in 2002, when the results of a randomized controlled trial showed the superiority of voriconazole over amphotericin B deoxycholate [6]. The greater clinical efficacy of voriconazole, coupled with a favorable safety profile and the availability of an oral formulation, cemented its position as the new standard of care for invasive pulmonary aspergillosis and most other pulmonary mold diseases, with the exception of mucormycosis [7,8,9]. In later clinical trials, posaconazole and isavuconazole showed similar efficacy compared to that of voriconazole, with fewer treatment-related adverse events [10,11]. However, the effectiveness of triazoles is undermined by the increasing occurrence of pulmonary mold diseases caused by fungal pathogens with acquired or intrinsic resistance to these drugs [12,13,14]. Strains of *Aspergillus fumigatus* with acquired resistance to azoles have been reported with increasing frequency since 1999 [13] and are currently found worldwide, with varying incidence rates [12]. Moreover, the widespread use of azoles as chemoprophylaxis in patients at high risk of invasive fungal disease, such as those undergoing remission-induction chemotherapy for acute leukemia or hematopoietic stem cell transplantation, implies that a growing proportion of fungal diseases in this population are diagnosed as breakthrough infections [15]. Controlled data on management of breakthrough fungal infections are lacking, as these patients were excluded from clinical trials of antifungal drug efficacy. Subject to this limitation, lipid formulations of amphotericin B are considered the preferred treatment of patients with suspected or proven infections with azole-resistant or breakthrough pulmonary mold infections [15]. Combined, these broad changes in fungal disease epidemiology increasingly constrain the treatment options available for pulmonary fungal diseases. Much needed drugs with novel antifungal mechanisms of action are currently undergoing evaluation in clinical trials [16]. Their roles in the treatment of pulmonary IFD remain to be determined.

This review will cover the pharmacology and antimicrobial spectrum of antifungal drugs used in the systemic treatment of pulmonary fungal diseases, highlighting the clinical trial data supporting targeted treatment of specific disease entities.

## 2. Triazoles

The triazoles, so named because of the three nitrogen atoms in their azole ring structure, were introduced into clinical use in the 1990s with the release of fluconazole. They have completely replaced imidazole antifungals for the systemic treatment of mycoses. The mechanism of action of azoles involves inhibition of lanosterol 14-alpha demethylase, a key enzyme in the ergosterol biosynthetic pathway. Depletion of cytoplasmic membrane ergosterol impairs the activity of membrane-associated enzymes and normal membrane function, thus, inhibiting fungal growth. Although the activity of azoles against *Candida* species is fungistatic, they are fungicidal against filamentous fungi. In vivo drug efficacy correlates with area under the concentration (AUC) curve over the MIC of the target organism (AUC/MIC) [17,18].

Five triazoles are currently licensed for systemic treatment of pulmonary fungal diseases: fluconazole, itraconazole, voriconazole, posaconazole and isavuconazole. Differences in affinity for 14-α demethylase account for their different spectrum of activity; fluconazole is active against common yeasts and endemic dimorphic fungi, voriconazole has an extended spectrum that includes *Aspergillus* and other pathogenic mold species, and posaconazole and isavuconazole have additional activity against mucorales. Cross-inhibition of human cytochrome P450 (CYP) enzymes causes clinically significant drug–drug interactions, with varying degrees among triazoles. Itraconazole, voriconazole and posaconazole are strong inhibitors of CYP3A4 and have the potential to increase serum levels of co-administered anticoagulants (warfarin, rivaroxaban), immunosuppressants (cyclosporine, tacrolimus, sirolimus, ruxolitinib, ibrutinib and vinca alkaloids), antiepileptics (carbamazepine, phenytoin) and many others. Extensive reviews of these interactions and guidance for their management are available [19,20].

### 2.1. Fluconazole

Fluconazole has ~90% bioavailability that is not affected by gastric acidity or food [21,22] and a long half-life, allowing for once-daily dosing. It is widely distributed in body compartments, reaching therapeutic concentrations in the CSF and ocular compartment. Two-thirds to three-quarters of the drug is excreted unchanged in the urine, requiring dose adjustment for renal insufficiency [23]. No adjustment is required for hepatic disease (Table 1).

#### 2.1.1. Spectrum of Activity

Fluconazole is active against most *Candida* species (which will not be discussed here as they are generally not considered pulmonary pathogens), Cryptococcus species and endemic dimorphic fungi, including *Coccidioides immitis*, *Histoplasma capsulatum* and *Blastomyces dermatitidis*. It has no clinically relevant activity against *Aspergillus* species and other pathogenic molds.

#### 2.1.2. Safety

Fluconazole is generally well tolerated, even when administered at high doses for extended periods, as used in the treatment of coccidioidomycosis. Frequent adverse events include headache, alopecia, xerosis and cheilitis, all reversible after discontinuation of therapy [24]. Hepatotoxicity and QTc prolongation may occur, as with other azoles. These effects are not dose related.

Fluconazole is a strong inhibitor of CYP2C19 and CYP2C9 and a moderate inhibitor of CYP3A4. Drugs that are metabolized through CYP3A4 and have the potential to prolong the QT interval, such as cisapride, astemizole, pimozide and quinidine, are contraindicated with fluconazole.

#### 2.1.3. Therapeutic Drug Monitoring

Because of the high bioavailability in fluconazoles, predictable linear pharmacokinetics and lack of defined therapeutic range, routine monitoring of blood levels is not recommended [25].

#### 2.1.4. Clinical Trials

##### Cryptococcosis See Discussion under “Amphotericin B”

**Coccidioidomycosis**. Fluconazole is considered the drug of choice for early uncomplicated coccidioidal pneumonia requiring treatment, including patients with significantly debilitating symptoms, extensive pneumonia, frailty and multiple comorbidities, in particular diabetes mellitus [26]. There are no randomized trials to support this recommendation. In fact, observational studies have not shown any clinical benefit of fluconazole treatment versus no treatment for patients with early coccidioidal pneumonia [27,28]. A randomized clinical trial compared fluconazole and itraconazole for the treatment of chronic cavitary coccidioidomycosis; the clinical response rate was marginally higher for itraconazole: 72% versus 57% after 12 months, *p* = 0.05 [29]. Both drugs are considered treatment options in for chronic pulmonary disease [26].

### 2.2. Itraconazole

Itraconazole is licensed for the treatment of allergic bronchopulmonary aspergillosis, chronic cavitary pulmonary aspergillosis and as second-line treatment for invasive pulmonary aspergillosis. It is also indicated for treatment of pulmonary coccidioidomycosis (in patients at risk of severe disease), histoplasmosis and blastomycosis (mild to moderate disease or step-down treatment after amphotericin B for more severe infection), cryptococcosis and sporotrichosis [30].

Itraconazole is available in three formulations: capsule, oral solution and IV formulation. In both the oral solution and IV formulation, itraconazole is complexed with sulfobutyl ether β-cyclodextrin sodium (SBECD). The oral solution is preferred over the capsules due to better bioavailability, lesser interpatient variability and 30% higher serum concentrations [31]. Itraconazole is highly lipophilic, and its complex with SBECD allows for aqueous formulation. Itraconazole requires an acidic pH to become soluble and, therefore, capsules must be taken after a meal for optimal absorption. Absorption is enhanced by acidic beverages and inhibited by proton pump inhibitors. In contrast, the oral solution is optimally absorbed in the fasting state. Itraconazole undergoes extensive first-pass metabolism in the liver, where it is transformed by the CYP450 isoenzyme 3A4 into hydroxy-itraconazole, a metabolite with antifungal activity [32]. Itraconazole is 99.8% bound to plasma proteins [32], reaching only negligible concentrations in CSF and ocular fluid, but high concentrations (~400 ng/mL) in bronchial fluid [32]. Studies have not shown a significant effect of renal impairment or hemodialysis on itraconazole pharmacokinetics [32]. The elimination half-life of itraconazole is longer in patients with cirrhosis, but the overall AUC is not affected [32]. Patients with liver impairment require careful monitoring for hepatotoxicity and may be at risk of drug–drug interactions over extended periods of time [30] (Table 1).

#### 2.2.1. Spectrum of Activity

Itraconazole′s spectrum of activity includes *Aspergillus* species, dimorphic fungi, including *C. immitis*, *H. capsulatum*, *B. dermatitidis* and *Paracoccidioides* species, and *Sporothrix schenckii*. Activity against *Fusarium* species and the Mucorales is minimal.

As of 2020, the European Committee on Antimicrobial Susceptibility Testing (EUCAST) defines antifungal clinical breakpoints for *Aspergillus* species [33]. The itraconazole clinical breakpoint for *A. fumigatus* and *A. flavus* is 1 mg/L [33]. Isolates with itraconazole MIC 2 mg/L are reported as resistant, with an accompanying statement indicating that voriconazole could be used for certain non-invasive infections if sufficient exposure is ensured. The Clinical Laboratory Standards Institute (CLSI) has not published clinical breakpoints for *Aspergillus* species.

#### 2.2.2. Safety

Frequent adverse effects of itraconazole include nausea, vomiting, abdominal discomfort and diarrhea. Together with voriconazole, itraconazole is the azole associated with the highest rates of hepatotoxicity [34,35]. Overall, 17% of patients treated with itraconazole experience elevated liver enzymes and 1.5% discontinue treatment because of liver toxicity [35]. Liver and renal toxicity was shown to be enhanced in patients receiving cyclophosphamide-based conditioning for stem cell transplantation because of inhibition of hepatic P-450 isoenzymes and increased exposure to toxic metabolites [36]. Adrenal insufficiency or mineralocorticoid excess manifesting as hypertension and hypokalemia with low plasma renin activity can occur at doses over 400 mg/day [37,38]. Itraconazole carries a black box warning for new or worsening heart failure, which may present as edema with reduced or preserved ejection fraction [39].

#### 2.2.3. Therapeutic Drug Monitoring

Serum concentrations should be monitored for all patients receiving itraconazole for systemic infections. Levels of 1–2 mcg/mL have been linked to treatment success [25]. Both itraconazole and hydroxy-itraconazole levels are reported and the sum is considered as the drug level. Because of the long half-life of itraconazole (~24 h), the timing of blood drug-level measurement does not seem to be important.

#### 2.2.4. Clinical Trials

**Aspergillosis**. Treatment of patients with corticosteroid-dependent ABPA with itraconazole was associated with significantly higher odds of clinical improvement, as compared with placebo [40]. Observational data indicate a 60% clinical response rate for patients with chronic pulmonary aspergillosis [41]. A randomized controlled trial compared treatment of chronic cavitary pulmonary aspergillosis with itraconazole (400 mg daily) for 6 months versus supportive treatment alone. Clinical and radiologic response was significantly more frequent in the itraconazole arm (76% versus 35%, respectively) [42]. Itraconazole is not a preferred agent for the treatment of invasive pulmonary aspergillosis, as its efficacy has not been rigorously compared to the current standard of care [43].

**Endemic mycoses**. Treatment of endemic mycoses with itraconazole has not been assessed in randomized controlled trials. According to current guidelines, itraconazole is used in the treatment of mild to moderate nonmeningeal blastomycosis and histoplasmosis [44,45]. Prospective observational data showed clinical improvement in 90% of patients with blastomycoses and 81% of patients with histoplasmosis [46].

### 2.3. Voriconazole

Voriconazole was the first of the second-generation triazoles to be approved by the US Food and Drug Administration (FDA). It is licensed for the treatment of invasive aspergillosis, as well as serious infections caused by *Scedosporium apiospermum* and *Fusarium* species, in patients intolerant of, or refractory to, other treatment [47]. Voriconazole is recommended as primary therapy of invasive pulmonary aspergillosis by the Infectious Diseases Society of America (IDSA) [7], the European Society for Microbiology and Infectious Diseases (ESCMID) [8] and the European Conference on Infections in Leukemia (ECIL) [9], based on the results of a seminal randomized controlled trial showing the superiority of voriconazole over amphotericin B deoxycholate [6].

Voriconazole is a synthetic derivative of fluconazole, with a fluorinated pyrimidine replacing one of the triazole rings and an added alpha-methyl group, which confer an extended spectrum of antifungal activity. These structural changes translate into higher avidity to the 14α-demethylase of *Aspergillus* species compared to that of Candida species. Moreover, while activity against *Candida* species is fungistatic, voriconazole and other second-generation azoles have potent fungicidal activity against *Aspergillus* species [48].

Voriconazole is available as oral and intravenous formulations. Intravenous voriconazole is solubilized in SBECD. The oral formulation has a bioavailability of >90% if ingested at least 1 h before or after a meal and ~80% if taken with a fatty meal. Steady-state plasma levels are achieved after 5–6 days without a loading dose, and after 1 day with a loading dose [49]. The pharmacokinetics of voriconazole in adults are non-linear [49]. Elimination is linear in children, and higher doses are required to achieve target plasma levels [50]. Around 20% of non-Indian Asians are poor metabolizers of voriconazole due to low CYP2C19 activity and are prone to adverse events from drug accumulation. The dose of voriconazole should be halved in patients with mild to moderate liver disease, and the drug should be avoided in patients with severe liver disease. For the oral formulation, no dose adjustment is necessary for renal dysfunction. Guidelines suggest that intravenous voriconazole should be avoided if creatinine clearance is below 50 mL/min, due to potential renal toxicity from accumulation of SBECD [47]. However, treatment of patients with baseline renal impairment for short periods (<10 days) was not associated with adverse outcomes in observational case series [51,52,53]. Thus, for patients for whom IV treatment is indicated and transition to oral therapy can be performed within 10 days, the risk–benefit ratio appears to be favorable (Table 1).

#### 2.3.1. Spectrum of Activity

Voriconazole has broad activity against filamentous fungi, including the major *Aspergillus* species complexes, dematiaceous molds, such as *Alternaria*, *Bipolaris* and *Exerohilum* species and *Scedosporium apiospermum*, hyaline molds, including *Fusarium*, *Talaromyces* and *Paecilomyces* species, and endemic dimorphic fungi. Importantly, voriconazole has no clinically relevant activity against members of the order Mucorales.

Certain cryptic Aspergillus species are intrinsically resistant to voriconazole, notably A. calidoustus, A. pseudodeflectus (section Usti), A. lentulus, A. felis, A. viridinutans, Neosartorya pseudofischeri, A. hiratsukae and A. tsurutae (section Fumigati). Acquired azole resistance in A. fumigatus is almost always associated with mutations in the CYP51A coding sequence, promotor or both. Mutations, such as TR34/L98H and TR46/Y121F/T289A, have been classified as environmental, meaning they are acquired from environmental exposure, typically by patients with no previous exposure to azole antifungals [54,55]. Other mutations, including amino acid substitutions in positions G54 and M200, have been found in patients with chronic pulmonary aspergillosis and are thought to arise as a result of long-term azole therapy [56,57].

Invasive aspergillosis caused by *A. fumigatus* with voriconazole MIC > 2 mg/L was associated with significantly greater mortality relative to isolates with lower MIC [58]. In line with these data, the EUCAST defines a clinical breakpoint of 1 mg/L for voriconazole and *A. fumigatus* [33]. Isolates with MIC 2 mg/L are reported as resistant, with an accompanying statement indicating that voriconazole could be used for certain non-invasive infections if sufficient exposure is ensured. *Fusarium* species are frequently associated with high voriconazole MICs. However, treatment of invasive fusariosis with voriconazole appears to be associated with lower mortality rates than those previously observed with amphotericin B deoxycholate, with no clear correlation with in vitro susceptibility testing results [59].

#### 2.3.2. Safety

Voriconazole is uniquely associated with photopsia, a reversible alteration in vision, which occurs in ~30% of treated patients. Changes in vision can include altered color discrimination, blurred vision, photophobia and the appearance of bright spots and wavy lines. Photopsia rarely requires discontinuation of voriconazole. Patients should be cautioned against driving. Visual hallucinations occur in ~15% of patients, often during receipt of intravenous loading doses, and are distinct from photopsia [60]. Patients may describe seeing unfamiliar people or creatures in their room or being in unfamiliar, surreal environments, while being fully aware that they are hallucinating. Visual and auditory hallucinations are associated with elevated voriconazole levels [60].

Other frequent adverse events include skin rash, photosensitivity and elevated liver enzymes. Hepatic injury and neurotoxicity are associated with elevated voriconazole serum levels [61]. Liver function tests should be checked at baseline, 2 weeks after starting therapy and every 2–4 weeks after that. Long-term treatment is associated with fluorosis leading to osteoarticular pain, periostitis and fractures [62]. Bone changes occur as a result of replacement of the hydroxyl ion in hydroxyapatite by fluoride. The resulting fluorapatite inhibits bone resorption and stimulates osteoblasts, leading to increased bone mineral density. Impaired renal function was noted in some patients with voriconazole-induced fluorosis [63]. Treatment with voriconazole is associated with an increased risk of cutaneous squamous cell carcinoma [64], warranting dermatologic surveillance.

#### 2.3.3. Therapeutic Drug Monitoring

Therapeutic monitoring of blood voriconazole levels is recommended for patients receiving voriconazole treatment because of significant interpatient pharmacokinetic variability [7,65]. Clinical response is associated with a minimum blood concentration of 1–1.5 mg/L or a trough concentration/MIC ratio of 2–5, whereas adverse events (neurotoxicity and hepatoxicity) increase in frequency at concentrations exceeding 4–6 mg/L. A higher therapeutic threshold should be considered for patients with invasive aspergillosis, disseminated disease or CNS involvement [65]. Voriconazole TDM was found to be associated with increased rates of clinical response and fewer drug discontinuations due to adverse events [66,67,68]. Voriconazole levels should be checked 5–7 days after starting treatment, changing the dose or switching from IV to oral treatment [65], considering the time required to achieve a steady state. Levels should also be tested in case of clinical worsening or non-response.

#### 2.3.4. Clinical Trials

**Invasive pulmonary aspergillosis**. Voriconazole became the standard of care for invasive aspergillosis following the results of a large randomized controlled trial comparing voriconazole and amphotericin B deoxycholate [6]. Most patients underwent hematopoietic stem cell transplantation or treatment for acute leukemia, and more than 80% had invasive pulmonary aspergillosis. Primary treatment with intravenous voriconazole for at least 7 days followed by oral voriconazole versus amphotericin B deoxycholate (1 to 1.5 mg/kg/day) was associated with significantly higher rates of complete or partial response at week 12 (53% versus 32%), with an absolute difference of 21.1% (95% confidence interval, 10.4% to 32.9%), meeting criteria for superiority of voriconazole over amphotericin B deoxycholate. Survival to week 12 was 71% versus 58% for voriconazole and amphotericin B deoxycholate, respectively (*p* = 0.02). Patients treated with voriconazole had significantly fewer drug-related adverse events. Voriconazole has never been compared directly with liposomal amphotericin B for the treatment of invasive aspergillosis in a randomized controlled trial.

**Chronic pulmonary aspergillosis**. Voriconazole was assessed in non-comparative cohort studies of chronic pulmonary aspergillosis, either as primary therapy or second-line treatment of patients failing or intolerant to itraconazole or amphotericin B [69,70,71,72]. Response was documented in 43% to 80% of patients, with good overall tolerability and infrequent discontinuation due to drug toxicity.

**Invasive fusariosis**. Randomized controlled trials are lacking for invasive fusariosis. However, observational data suggest that treatment of invasive fusariosis with voriconazole is associated with higher overall survival rates than treatment with amphotericin B deoxycholate [73]. Analysis of 233 cases from 11 countries showed that the 90-day survival rate increased from 22% in the period between 1985 and 2000 to 43% from 2001 to 2011, corresponding to a shift from amphotericin B to voriconazole treatment. The survival rate was 60% with voriconazole, 53% with lipid formulations of amphotericin B and 28% with amphotericin B deoxycholate (*p* = 0.04) [73].

**Scedosporiosis**. The efficacy of voriconazole for the treatment of scedosporiosis was assessed in a retrospective trial that included 107 patients, 24% of whom had pulmonary or sinus infections [74]. Underlying conditions were solid organ transplantation (22%), hematologic malignancy (21%) and surgery or trauma (15%). A successful therapeutic response was achieved for 57% of patients. Forty percent of patients died (median survival time, 133 days). The survival rate was significantly greater for patients infected with *S. apiospermum* versus *S. prolificans*.

### 2.4. Posaconazole

Posaconazole is structurally similar to itraconazole, but contains two fluorine molecules in place of chlorine and a furan ring in place of the dioxolane ring. There are three formulations: oral suspension (40 mg/mL), delayed-release (DR) tablet (100 mg) and intravenous. The DR tablets are preferred over oral suspensions because of better tolerability and more predictable absorption. Posaconazole is approved by the FDA for second-line treatment of invasive aspergillosis and chronic cavitary pulmonary aspergillosis, as well as prophylaxis of invasive fungal diseases in patients with hematologic malignancies and prolonged neutropenia or hematopoietic stem cell transplantation (HSCT) with graft-versus-host disease (GVHD) [75].

The oral suspension formulation of posaconazole has saturable absorption and variable bioavailability, requiring 3- or 4-times daily administration. Bioavailability is affected by food intake, gastric pH and motility; serum levels are significantly higher if taken with a high-fat meal [76], and subtherapeutic levels may be found in patients with limited food intake or concomitant use of H2-receptor blockers or metoclopramide [77].

The delayed-release formulation is a pH-sensitive film-coated tablet designed to limit posaconazole release in the low-pH gastric environment, instead releasing the drug in the neutral pH of the small intestine. Because of this, food intake and gastric pH have marginal effects on serum posaconazole concentrations when taking the DR formulation [78,79]. Moreover, drug exposure (area under the serum concentration curve) is increased three-fold with the DR formulation relative to the oral suspension in the fasting state, and blood levels are 35% higher in the fed state [80]. Other advantages of the DR formulation over the oral suspension are its linear pharmacokinetics and once-daily administration [81]. Nevertheless, highly variable blood levels were observed among patients with hematologic malignancies receiving high-dose oral DR posaconazole [82]. The intravenous formulation is administered once daily at doses identical to the DR tablet. Posaconazole is >98% bound to plasma proteins. Most of the drug is eliminated unchanged in feces, with minimal renal clearance. No dose adjustment is required for oral or IV formulations in patients with renal or hepatic-function impairment. The IV formulation contains SBECD, which may accumulate in patients with creatinine clearance < 50 mL/min. As noted above, a nephrotoxic effect of SBECD has not been observed in patients during short courses of treatment [51,52,53]. Posaconazole is non-dialyzable, and no dose adjustment is recommended during dialysis. It is highly lipophilic and achieves high concentrations in the lungs, kidneys, liver and heart; specifically, posaconazole concentrations in alveolar cells are about 32-fold higher than blood concentrations [83]. Penetration into the CSF is variable, ranging from undetectable to 237% serum levels [84,85,86] (Table 1).

#### 2.4.1. Spectrum of Activity

The antifungal spectrum of posaconazole is similar to that of voriconazole, with the notable addition of activity against Mucorales. Compared with voriconazole, posaconazole has a lower MIC against *A. fumigatus* and higher MIC against A. niger [87]. The clinical consequences of these differences are unknown.

The EUCAST posaconazole clinical breakpoint for *A. fumigatus* is 0.25 mg/L. Isolates with a posaconazole MIC of 0.25 are defined as susceptible if they are also susceptible to itraconazole [33].

#### 2.4.2. Safety

Adverse effects are less frequent with posaconazole than with itraconazole or voriconazole [10]. Gastrointestinal effects, such as nausea, vomiting and diarrhea, are more common with the oral suspension than the DR tablet. Hepatotoxicity and QTc prolongation have also been observed. Hypertension, hypokalemia and peripheral edema have been attributed to mineralocorticoid excess due to inhibition of adrenal 11-β hydroxylase and peripheral 11-β hydroxysteroid dehydrogenase by posaconazole [88].

Posaconazole is a weaker inhibitor of CYP P450 enzymes than itraconazole or voriconazole. In addition, posaconazole inhibits CYP 3A4 and p-glycoprotein, but unlike itraconazole and voriconazole, it lacks significant activity on CYP 2C9 and CYP C19 [20]. It is a substrate and inhibitor of the P-glycoprotein transport system [20].

#### 2.4.3. Therapeutic Drug Monitoring

Clinical effectiveness of posaconazole for the treatment of invasive aspergillosis is associated with achievement of serum levels ≥ 1 mg/L [89]. Higher concentrations may be required for the treatment of Mucorales, Fusarium and Scedosporium species. A concentration threshold for drug toxicity has not been established. TDM is recommended when using the oral suspension, treating non-Aspergillus molds and when clinical failure is suspected [25]. Trough levels should be measured at steady state, ≥7 days after treatment initiation or dose adjustment [25].

#### 2.4.4. Clinical Trials

Posaconazole was evaluated as prophylaxis in the setting of remission-induction chemotherapy for acute myelocytic leukemia and myelodysplastic syndrome (AML/MDS) [90] and HSCT with graft-versus-host disease [91]. Compared with fluconazole and itraconazole, preventive treatment with posaconazole was associated with significantly fewer invasive fungal diseases (specifically, invasive aspergillosis) and lower overall mortality in the AML/MDS study [90]. Posaconazole was more effective than fluconazole for the prevention of invasive aspergillosis and death attributed to fungal disease in HSCT recipients with GVHD. However, overall mortality was similar between study groups [91].

**Invasive aspergillosis**. Posaconazole (IV or DR tablet) was compared to voriconazole (IV or oral) for the treatment of invasive aspergillosis in a randomized controlled trial that included 653 participants [10]. Occurrence of the primary outcome (overall mortality in the intention-to-treat population at day 42) was 15% and 21% in the posaconazole and voriconazole groups, respectively. The adjusted treatment difference of −5.3% (95% confidence interval (CI), −11.6 to 1.0) met the predefined 10% non-inferiority margin [10]. The incidence of treatment-related adverse events, including those leading to drug discontinuation, was significantly lower with posaconazole versus voriconazole. Visual and neuropsychiatric events were more common with voriconazole, whereas hypokalemia and reduced appetite were more frequently associated with posaconazole.

**Chronic pulmonary aspergillosis (CPA)**. A retrospective cohort study of 79 patients with CPA treated with posaconazole oral suspension (400 mg twice daily) demonstrated a response rate of 61% at 6 months and 46% at 12 months [92]. For some patients, response occurred only after ≥1 year of treatment. Fifteen percent of patients experienced adverse events and nine discontinued treatment [92].

**Mucormycosis**. Posaconazole’s good in vitro activity against Mucorales and oral bioavailability make it an attractive option for treatment of patients with mucormycosis. A review of 96 cases of mucormycosis treated with posaconazole included 11 patients (11.5%) with pulmonary disease. Most patients received posaconazole as part of a first-line regimen, most frequently in combination with a lipid-based amphotericin B formulation [93].

**Invasive fusariosis**. Retrospective analysis of three open-label clinical trials identified 21 patients with invasive fusariosis (7 with pulmonary involvement) who received posaconazole as salvage therapy [94]. Treatment was successful in 48% of cases. The success rate was significantly lower for patients with persistent non-resolving neutropenia than for patients who recovered from neutropenia (20% versus 67%, respectively) [94].

### 2.5. Isavuconazole

Isavuconazole is the newest triazole in clinical use. It was approved by the FDA and EMA for the treatment of invasive aspergillosis and mucormycosis (for the latter, the EMA approval is for cases where amphotericin B is not appropriate) [95,96]. Isavuconazole is formulated for intravenous and oral use as isavuconazonium sulfate, a prodrug that is rapidly converted by plasma esterases to isavuconazole, the bioactive molecule. Oral capsules contain 186 mg isavuconazonium sulfate (100 mg isavuconazole), and powder for intravenous solution contains 372 mg isavuconazonium sulfate (200 mg isavuconazole) per dose. Dosing is identical for oral and intravenous formulations (Table 1). Because isavuconazonium sulfate is water-soluble, the intravenous formulation does not contain SBECD, and no adjustment is required for renal function. The half-life (184 h) is significantly longer than that of voriconazole or posaconazole, allowing for once-daily dosing. The bioavailability of the oral formulation is >97% and is not affected by food intake or gastric pH [97]. Pharmacokinetics are linear and more predictable than those of voriconazole. The volume of distribution is large, consistent with penetration into most tissues, including the central nervous system (Table 1).

#### 2.5.1. Spectrum of Activity

The in vitro activity of isavuconazole is generally similar to that of voriconazole, with some notable exceptions. Importantly, isavuconazole has clinically relevant activity against members of the order Mucorales, including *Rhizopus arrhizus*, the species most frequently isolated from patients with mucormycosis, as well as *Lichtheimia* and *Rhizomucor* species [98]. Activity is weaker against *Mucor circinelloides* and other *Mucor* species [99,100,101]. *Aspergillus* species with acquired resistance to voriconazole resulting from mutations in the CYP51A gene exhibit high-level resistance to isavuconazole [102]. Similarly, intrinsically resistant cryptic *Aspergillus* species, such as section Fumigati species *A. lentulus* and *A. udagawae* and section Usti member *A. calidoustus*, frequently exhibit pan-azole resistance [103,104,105]. In vitro activity against *Fusarium* species is limited, with MIC50 > 16 mg/L and no significant differences among species complexes [106].

New EUCAST guidance defines isavuconazole clinical breakpoints of 2 mg/L for *A. fumigatus* and *A. flavus*. Isolates with isavuconazole MIC 2 mg/L that are wild-type voriconazole are reported as isavuconazole-susceptible [33]. However, analysis of SECURE and VITAL trial data did not show a correlation between the MIC and treatment success for MIC < 16 mg/L [107].

#### 2.5.2. Safety

Isavuconazole has an overall favorable safety profile. In the SECURE clinical trial, treatment-related adverse events were less common with isavuconazole than with voriconazole (30% versus 40%, respectively), and the frequency of liver-, skin-/soft-tissue- and eye-related adverse events was significantly lower with isavuconazole [11]. Unlike other azoles, isavuconazole shortens rather than lengthens the QTc [108]. No cardiac risk has so far been associated with this effect, but isavuconazole should be avoided in patients with short QT syndrome and associated arrhythmia [109]. Caution and monitoring are advised in patients receiving drugs that shorten QTc, such as lamotrigine [109]. The clinical utility of isavuconazole’s improved safety profile was further confirmed by real-world observational data, showing that isavuconazole was well-tolerated in patients who discontinued posaconazole due to liver toxicity or QTc prolongation [82].

Isavuconazole is a moderate CYP3A4 inhibitor and, therefore, has milder effects on CYP3A4 substrate exposure than voriconazole, posaconazole and itraconazole. Importantly, exposure to immunosuppressive drugs (AUC0-∞), including cyclosporine, mycophenolic acid, sirolimus and tacrolimus, is increased by isavuconazole [110]. Monitoring of the plasma levels of these drugs with dose adjustment is mandatory during co-administration. Isavuconazole is also a mild inducer of CYP2B6 and has no significant effect on the activities of other CYP enzymes, including CYP2C19 [111]. Like other azoles, isavuconazole is metabolized by CYP3A4, and co-administration with potent CYP3A4 inducers profoundly reduces isavuconazole exposure. Specifically, rifampicin, the anticonvulsants phenytoin, carbamazepine and phenobarbital, and ritonavir lower the isavuconazole area under the concentration curve (AUC0-∞) by >90% [110], and their concomitant use is contraindicated.

#### 2.5.3. Therapeutic Drug Monitoring

TDM is not routinely required for isavuconazole, and its use has not been associated with improved clinical outcomes [112]. Further, pharmacokinetic modeling based on clinical trial data predicts that the currently recommended dosing regimen would adequately treat > 90% of infections with Aspergillus species with MIC of up to and including 1 mg/L by EUCAST methodology or 0.5 mg/L by CLSI methodology [113]. It is notable, however, that clinical failure rates were higher in obese patients enrolled in a clinical study of invasive candidiasis [114], suggesting that this group may be at risk of subtherapeutic levels. Moreover, an exposure difference of 40% was identified between Caucasian and Asian patients, the mechanism of which is unclear [113]. Other populations for which limited PK data are available are children and patients treated with renal replacement therapy or extracorporeal membrane oxygenation (ECMO) [115,116].

#### 2.5.4. Clinical Trials

**Invasive aspergillosis**. The SECURE trial was a phase 3 double-blinded randomized controlled clinical trial that aimed to assess the efficacy of isavuconazole versus voriconazole for the treatment of invasive aspergillosis [11]. Isavuconazole was started as intravenous infusion, with an option for oral switch from day 3. The primary study outcome of all-cause mortality through day 42 in the intention-to-treat population occurred in 19% of the isavuconazole group versus 20% of the voriconazole group. The adjusted treatment difference of −1.0% (95% CI, −7.8 to 5.7) satisfied the pre-defined non-inferiority margin of 10% [11]. The efficacy of both drugs was significantly lower in patients with unresolved versus resolved neutropenia [117]. Drug-related adverse events were reported in significantly fewer of the patients treated with isavuconazole versus voriconazole. A lower frequency of adverse events related to the hepatobiliary system, eyes and skin/soft tissue was observed in the isavuconazole group [11].

**Mucormycosis**. The effectiveness of isavuconazole for the treatment of mucormycosis was assessed in the open-label VITAL trial [118]. The study included 37 patients with mucormycosis, 22 of whom had pulmonary disease, either isolated (10 patients) or with extra-pulmonary involvement (12 patients). Isavuconazole was used as primary treatment (21 patients) or second-line treatment (16 patients, 11 with refractory disease and 5 intolerant to other antifungals). Treatment with isavuconazole was often prolonged, with a median treatment of 84 days (range, 2 to 882 days). Disease status, assessed by an independent data review committee, was: complete response, 0 patients (0%), partial response, 4 patients (11%), stable disease, 16 (43%), progression of disease, 1 (3%) and death, 13 (35%). The response rate (complete or partial) at the end of isavuconazole treatment was 32% for primary treatment and 36% for second-line treatment. The crude overall mortality of patients receiving primary isavuconazole treatment (33%) was similar to that of matched historical controls from the Fungiscope registry treated with amphotericin B (39%). These findings showed that isavuconazole could be used as primary or salvage treatment of mucormycosis, with efficacy comparable to that of amphotericin B as primary treatment of mucormycosis, and a favorable safety profile. Additional fungal diseases included in the VITAL population and their respective response rates to isavuconazole were cryptococcosis (*n* = 9, 60%), histoplasmosis (*n* = 7, 86%), coccidiodomycosis (*n* = 9, 56%) and paracoccidiodomycosis (*n* = 10, 80%) [118].

## 3. Amphotericin B Deoxycholate and Lipid Formulations

Amphotericin B (AMB) is a polyene antifungal with broad activity against yeasts and molds. It exerts rapid, concentration-dependent fungicidal activity by binding to ergosterol in the fungal cytoplasmic membrane. Amphotericin B inserts into the cell membrane and forms oligomers that act as channels, allowing for efflux of potassium, magnesium and organic substrates, causing rapid cell death. Recent studies have shown that binding to ergosterol is sufficient to induce cell death, irrespective of membrane pore formation [119]. AMB aggregates around membrane ergosterol and acts as a sponge, extracting ergosterol and sequestering it [120].

Amphotericin B deoxycholate (AMBD) was introduced in 1959 and, for many years, was the only available systemic antifungal. AMB is amphipathic and insoluble in water. AMBD can be dissolved in water to form a colloid solution. Only water or dextrose in water should be used to dissolve the powder—dissolution in electrolyte containing solutions results in colloid aggregates. In serum, AMB dissociates from deoxycholate and binds to serum proteins, chiefly beta lipoprotein, the carrier protein for cholesterol, albumin and erythrocytes [121]. AMB rapidly leaves the circulation and becomes associated with cholesterol in membranes of the liver and other organs. From these reservoirs, it is slowly released back into the circulation.

AMB is degraded in situ, with only a small fraction excreted in urine or bile. Blood concentrations are not affected by renal or hepatic function or dialysis. The concentration of AMB in inflamed pleural space, peritoneum, synovial fluid and vitreous humor is about two-thirds that of serum concentration. Penetration into bronchial secretions and the CNS is poor (Table 1).

### 3.1. Spectrum of Activity

Amphotericin B is active against most fungi, including the most important yeast and filamentous pathogens. Notable causes of pulmonary disease with intrinsic resistance to AMB include *Aspergillus terreus*, *Scedosporium* spp. and *Trichosporon* spp. Emergence of resistance to AMB is rare, possibly due to the significant fitness tradeoff associated with AMB resistance [122]. EUCAST defines an AMB clinical breakpoint of 1 mg/L for *A. fumigatus* and *A. niger* [33], consistent with data showing a significant mortality rate of invasive aspergillosis treated with AMBD and MIC ≥ 2 mg/L [123].

### 3.2. Safety

Toxicity associated with AMBD includes nephrotoxicity and infusion-related toxicity. Dose-dependent nephrotoxicity is caused by the vasoconstriction of afferent arterioles, which reduces glomerular and tubular perfusion. Irreversible renal damage occurs as a result of nephron loss due to ischemia and is associated with the total dose of AMBD administered. Nephrotoxicity is potentiated by co-administration with cyclosporine or aminoglycosides, which act similarly to AMB by constricting efferent arterioles, as well as by hypotension and volume depletion [124]. Administration of 500 to 1000 mL of normal saline immediately before or after AMB infusion (“sodium loading”) can reduce arteriolar vasoconstriction and help preserve GFR [125]. AMBD causes renal tubular acidosis, resulting in potassium, magnesium and bicarbonate wasting. Potassium levels should be carefully monitored and corrected during treatment.

Infusion reactions occur acutely during drug infusion and include fever, chills, tachypnea and hypoxemia. These reactions tend to be most severe with the first AMBD infusion and milder with each subsequent dose. Young children are less susceptible to infusion reactions than adults. Acute reactions can be treated with meperidine, which should be given with an antiemetic. A test dose (1 mg over 15 min) is used in some centers to prevent severe reactions in unstable patients.

### 3.3. Lipid Formulations

Lipid formulations of amphotericin B were developed to address the dose-limiting toxicity of this drug. Three such formulations were devised: AMB colloidal dispersion (ABCD), AMB lipid complex (ABLC) and liposomal AMB (LAMB). Liposomal AMB is by far the most widely used and studied of these formulations. In LAMB, AMB is incorporated into spherical vesicles composed of lipid bilayers. AMB is released from liposomes upon contact with fungal cells [126]. Envelopment of AMB in liposomes reduces the drug’s toxicity without affecting its activity. Following intravenous infusion, LAMB is distributed unevenly in organs, with the greatest concentrations found in the spleen and liver [127], likely reflecting uptake of liposomes by macrophages in these organs. LAMB achieves 5-times greater concentrations in pulmonary epithelial lining fluid compared with AMBD [128]. LAMB is cleared slowly from tissue, with a terminal half-life of 152 h. Urinary and biliary clearance of LAMB is significantly lower than that of AMBD [126]. Neither LAMB nor AMBD requires dose adjustment for creatinine clearance.

LAMB is the least nephrotoxic of the AMB formulations. A meta-analysis of five published clinical trials found a pooled nephrotoxicity rate of 32% with AMBD versus 14% with LAMB [129]. Infusion reactions are less frequent with LAMB than with AMBD or other AMB lipid formulations. Conversely, hepatotoxicity is more frequent with LAMB than with AMBD [130].

In ABLC, AMB is complexed with dimyristoylphosphatidylcholine and dimyristoylphosphatidylglycerol to form ribbon-like particles, whereas in ABCD, it is formulated with cholesteryl sulphate, forming disc-like particles. Both ABCD and ABLC are associated with lower rates of nephrotoxicity compared to AMBD. The rate of infusion-related toxicity is lower with ABLC than with AMBD. Moreover, ABLC appears to be well tolerated by patients who experienced an infusion reaction with LAMB [131]. However, the frequency of infusion reactions with ABCD is at least as high as with AMBD.

### 3.4. Clinical Trials

**Invasive aspergillosis**. AMBD was replaced by voriconazole as the drug of choice for the treatment of invasive aspergillosis following the definitive results of a trial showing the greater efficacy and safety of voriconazole [6]. However, LAMB has never been compared with voriconazole in a randomized controlled trial. The superior tolerability of LAMB versus AMBD, which allows for dosing at 5 mg/kg/day and higher for extended periods of time, might be advantageous for the treatment of invasive aspergillosis. In the Ambiload trial, patients with invasive mold infections were randomized to receive a 14-day LAMB loading dose of either 3 mg/kg/day (standard dose) or 10 mg/kg/day (high dose) [132]. In both arms, loading dose was followed by open-label LAMB at 3 mg/kg/day. Two hundred and one patients were recruited, 97% with invasive aspergillosis. Only 66% and 50% completed 14 days of treatment in the standard and high-dose groups, respectively. Most study discontinuations were due to drug-related adverse events. Seventy percent of patients were switched over to other antifungals, most commonly voriconazole and caspofungin. Favorable response at the end of study drug treatment was observed in 50% and 46% of standard and high-dose groups (*p* = 0.65). Overall survival to the end of study drug treatment was similar between study groups. There was a trend for higher overall survival in the standard-dose group compared with the high-dose group at 12 weeks (71% versus 58%, respectively, *p* = 0.089). However, Kaplan–Meier survival curves diverged after >14 days, suggesting that factors other than study arm were associated with this outcome. High-dose LAMB was associated with a significantly higher frequency of nephrotoxicity (31% versus 14% for standard- and high-dose groups, respectively; *p* < 0.01) and grade 3 hypokalemia (30% versus 16%, respectively; *p* = 0.015). Drug discontinuation due to adverse events was more common in the high-dose group (32% versus 20%, respectively; *p* = 0.035). In sum, the Ambiload trial showed that higher-than-standard doses of LAMB offered no benefit for the treatment of invasive aspergillosis and were potentially harmful. Indirectly, treatment outcomes observed for LAMB were similar to those seen with voriconazole in the randomized trial [6]. Thus, LAMB could be considered a reasonable alternative to voriconazole for patients with invasive aspergillosis who are either refractory or intolerant to triazoles [7,8,9].

**Mucormycosis**. Lipid formulations of AMB are considered the mainstay treatment of mucormycosis, based on uncontrolled case series and substantial clinical experience [133]. Randomized controlled trials have not been conducted, and the prospects for such trials in the future are limited by the rarity and lethality of this infection, as well as the need to account for multiple confounding underlying conditions and interventions. Analysis of 230 cases of mucormycosis collected from 13 European countries (30% with pulmonary involvement) revealed that treatment with AMB was significantly associated with survival, together with trauma as an underlying condition, surgical treatment and younger age [134]. In a series of 60 prospectively collected cases from Italy that included 15 cases of pulmonary infection [135], LAMB was the most frequent antifungal treatment (44 patients; median dose, 5 mg/kg/day) and was significantly associated with the likelihood of survival. A prospective pilot clinical trial of high-dose LAMB (10 mg/kg) enrolled 40 patients with mucormycosis, including 10 with pulmonary involvement [136]. Favorable response was documented in 12 patients (36%) after 4 weeks and in 14 patients (45%) after 12 weeks. Doubling of serum creatinine occurred in 40% of patients. Taken together, observational data support the use of LAMB or ABLC at a minimal dose of 5 mg/kg/day for the treatment of mucormycosis [133]. Studies in diabetic mice have shown that a LAMB dose of 7.5 mg/kg/day and an ABLC dose of 15 mg/kg/day are required to treat CNS mucormycosis [137]. The European Confederation of Medical, Mycology (ECMM) clinical guidelines recommend LAMB at a dose of 10 mg/kg/day if the CNS is involved [133].

**Cryptococcosis**. AMBD has been studied extensively for the treatment of cryptococcal meningitis [138,139,140]. Initial induction therapy with AMBD (1 mg/kg/day) in combination with flucytosine improves the survival of patients with cryptococcal meningitis and results in more rapid fungal clearance from the CSF compared with AMBD alone or in combination with fluconazole [138,139,141]. The Advancing Cryptococcal Meningitis Treatment for Africa (ACTA) trial demonstrated equivalent efficacy for 7 days of AMBD plus flucytosine, 14 days of AMBD plus flucytosine and 14 days of an all-oral regimen of fluconazole and flucytosine [138]. A single high dose of LAMB (10 mg/kg) followed by an oral regimen of fluconazole and flucytosine for 14 days was non-inferior to 7 days of AMBD plus flucytosine with fewer renal and hematologic adverse events [142] and is currently the preferred regimen recommended by the WHO [143]. There have been no controlled clinical trials specifically assessing the treatment of cryptococcal pulmonary disease. In immunosuppressed individuals, CNS and disseminated infection should be excluded by clinical evaluation, as well as culture and cryptococcal antigen testing of blood and CSF. Patients with combined pulmonary and CNS infection are treated with combination regimens appropriate for cryptococcal meningitis [140]. Isolated, mild to moderate pneumonia may be managed with fluconazole monotherapy. However, severe pneumonia (e.g., presenting with acute respiratory distress syndrome) should be treated with a regimen that includes an AMB formulation and flucytosine [140].

**Endemic fungi**. Observational data showed higher rates of survival among patients with disseminated histoplasmosis treated with AMBD than among those not treated [144]. A randomized clinical trial of patients with moderate to severe disseminated histoplasmosis found a significantly higher clinical success rate with LAMB compared to AMBD (88% versus 64%), with fewer adverse events in the LAMB arm [145]. Among 84 patients with endemic mycoses included in the CLEAR registry, treatment with ABLC was associated with a favorable outcome in 76% [146].

## 4. Echinocandins

The echinocandins are members of the lipopeptide class of antifungal drugs, composed of cyclic hexapeptides N-linked to a fatty acyl side chain [147]. They non-competitively inhibit beta-1,3-D-glucan synthase, interfering with biosynthesis of beta-1,3-D-glucan, a polysaccharide in the cell wall of many fungi. This results in fungicidal activity against *Candida* species and a more subtle inhibition of normal growth of *Aspergillus* species. The latter manifests as morphologic change in the mycelium due to slowed growth and lysis of hyphal tips [148,149]. There are currently three approved echinocandins: caspofungin, micafungin and anidulafungin. Rezafungin is a novel echinocandin with a long elimination half-life (~133 h), allowing for once-weekly dosing. Recently published results of a phase 3 clinical trial have shown non-inferiority compared with caspofungin for the treatment of candidemia and invasive candidiasis [150]. Echinocandins are recommended by IDSA and ESCMID guidelines as first-line treatment of invasive candidiasis and candidemia [151,152,153]. Support for their use in the treatment of invasive aspergillosis is limited, and they are considered as salvage therapy when first-line agents have failed or in situations where azoles and amphotericin B formulations are contraindicated [7].

Echinocandins are available for intravenous use only. All three approved echinocandins have plasma half-lives of >10 h, compatible with once-daily dosing. Caspofungin and micafungin are >95% bound to plasma proteins, whereas anidulafungin is 84% protein bound [147]. Echinocandins reach therapeutic concentrations in most organs, including the lungs, liver and spleen, but penetrate poorly into the eye and uninfected CSF. Caspofungin and micafungin undergo degradation in the liver by hydrolysis and N-acetylation, followed by prolonged excretion in the bile [147]. Anidulafungin undergoes spontaneous chemical degradation with fragment elimination in bile. The concentrations of all echinocandins in the urine are negligible, and no dose adjustment is required for renal impairment. Echinocandins are non-dialyzable and require no dose adjustment in patients undergoing dialysis. The prescribing information for caspofungin recommends reducing the maintenance dose by 50% for patients with moderate hepatic impairment [154]. However, studies showing only marginal effects of Child B or C cirrhosis on caspofungin pharmacokinetics have called this recommendation into question [155]. No hepatic dose adjustment is required for other echinocandins (Table 1).

### 4.1. Safety

Echinocandins have a favorable safety profile. Elevated liver enzymes, ALT, AST or alkaline phosphatase, occur in ~13% of patients. Coadministration of caspofungin and cyclosporine increases the risk of hepatotoxicity and should be avoided. Anidulafungin appears to be associated with a lower risk of hepatotoxicity than other echinocandins, and is well tolerated by patients who developed hepatotoxicity with caspofungin treatment [156]. Drug interactions were not observed with micafungin or anidulafungin. Anaphylactoid reactions (flushing, rash, facial swelling, hypotension and bronchospasm) are attributed to rapid infusion triggering histamine release [154].

### 4.2. Clinical Trials

**Invasive aspergillosis**. Support for the use of echinocandins as primary or salvage therapy for invasive aspergillosis comes from relatively small cohort studies. A non-comparative trial studied caspofungin as primary therapy for 61 patients with invasive aspergillosis (60 with pulmonary disease) [157]. Complete or partial response was observed in 33% of patients. Caspofungin was studied in a small cohort of hematopoietic stem cell transplantation recipients [158]. Thirty-three percent of patients had complete or partial response to treatment after 12 weeks. In a dose-ranging study, caspofungin administered at daily doses of up to 200 mg was well tolerated and associated with a response rate of 54.3% [159]. Among 225 patients with invasive pulmonary aspergillosis, response to primary or salvage therapy with micafungin (monotherapy, 34 patients, combination with other agents, 191 patients) was observed in 35.6% of patients [160]. Favorable response was documented in 45% of 83 patients with invasive aspergillosis and failure of, or intolerance to, first-line therapy who were treated with caspofungin, versus 16% of historical controls, most of whom were treated with amphotericin B and/or itraconazole [161].

Combination therapy with voriconazole and anidulafungin was compared with voriconazole monotherapy in a randomized clinical trial that included 454 patients with invasive aspergillosis [162]. The 6-week mortality rate (the primary outcome) was 19.3% in the combination arm and 27.5% in the monotherapy arm (difference, −8.2 percentage points (95% CI, −19.0 to 1.5); *p* = 0.087). Post hoc analysis revealed a significant survival benefit with combination therapy for a subset of patients with positive serum galactomannan. The serum galactomannan index correlates with the risk of death from invasive aspergillosis and may be an indirect measure of fungal burden and angioinvasion [163,164,165]; thus, this finding might suggest a potential target population that could benefit from primary combination treatment. However, the study was not adequately powered to detect a treatment effect, and combination therapy is not currently recommended as a standard of care for invasive aspergillosis [7]. Data from small observational studies suggest that echinocandins in combination with voriconazole may be effective as salvage treatment for invasive aspergillosis [166,167]. These results have yet to be confirmed by adequately controlled clinical trials.

**Chronic pulmonary aspergillosis**. Treatment of CPA with intravenous echinocandins is typically considered in cases of disease progression, intolerance or resistance to primary azole treatment. In a randomized controlled study, micafungin was non-inferior to IV voriconazole for the treatment of CPA [168]. Fewer adverse events occurred with micafungin versus voriconazole. The efficacy and safety profiles of caspofungin were similar to those of micafungin [169]. Long-term treatment of CPA with echinocandins is impractical given the requirement for intravenous administration. Some have suggested cyclic treatment with an echinocandin with azole treatment in between doses [170].

## 5. Summary

The last two decades have seen significant progress in the field of systemic antifungal therapeutics. Landmarks include the replacement of AMBD with voriconazole for the treatment of invasive aspergillosis and most other invasive mold diseases, the introduction of echinocandins, a new class of systemic antifungals and significant gains in knowledge about the optimal treatment of cryptococcosis. However, the range of antifungal classes available for systemic treatment remains limited, and resistance to antifungals, both intrinsic and acquired, is a growing healthcare menace. Highly anticipated antifungals representing novel drug classes are currently undergoing clinical trials [171,172].

## Figures and Tables

**Table 1 jof-09-00144-t001:** Characteristics of systemic antifungal drugs.

Drug	Bioavailability	Dose	Renal Adjust	Hepatic Adjust	Adverse Effects	TDM
Fluconazole	90%	Pulmonary coccidioidomycosis: 400 mg to 800 mg daily	CrCl ≤ 50 mL/min: reduce by 50%	None. Monitor for toxicity	Hepatotoxicity, QTc prolongation, headache, alopecia, xerosis, cheilitis	No
Itraconazole	Capsule: 55% Oral solution: 80%	200 mg 1 to 3 times daily	None	None. Monitor for toxicity	Abdominal pain, Nausea, vomiting Hepatotoxicity Hypertension, hypokalemia Heart failure QTc prolongation	Yes
Voriconazole	96%	IV: 6 mg/kg twice daily for 2 doses, then 4 mg/kg twice daily Oral: 200 to 300 mg twice daily or weight-based dosing (3 to 4 mg/kg twice daily)	None CrCl ≤ 50 mL/min: Consider IV to oral switch or alternative agent	Mild to moderate impairment: 50% dose Severe impairment: consider benefit vs. risk, monitor toxicity	Abdominal pain, Nausea, vomiting Hepatotoxicity Neurotoxicity (visual hallucinations) Photopsia Photosensitivity Hypertension QTc prolongation	Yes
Posaconazole	50%	Oral suspension: 200 mg 4 times daily DR tablet and IV: 300 mg twice daily for 2 doses, then 300 mg once daily	None CrCl ≤ 50 mL/min: Consider IV to oral switch or alternative agent	None. Monitor for toxicity	Abdominal pain, Nausea, vomiting Hepatotoxicity Hypertension, hypokalemia QTc prolongation	Yes
Isavuconazole	>97%	Oral and IV: 200 mg 3 times daily for 48 h, then 200 mg daily	None	None. Monitor for toxicity	QTc shortening	No
Caspofungin	NA	Loading dose: 70 mg IV Maintenance dose: 50 mg IV once daily	None	Moderate liver impairment: reduce dose by 50%	Hypotension, edema, diarrhea, nausea, vomiting, chills, headache, rash, hepatotoxicity	No
Micafungin	NA	100 mg to 150 mg once daily IV	None	None		No
Anidulafungin	NA	Loading dose: 200 mg IV Maintenance dose: 100 mg IV once daily	None	None		No
Amphotericin B deoxycholate	NA	Invasive aspergillosis: 1–1.5 mg/kg/day Cryptococcosis (meningoencephalitis, severe pneumonia), histoplasmosis, blastomycosis: 0.7–1 mg/kg/day	None	None	Infusion related toxicity (fever, chills, hypoxemia), nephrotoxicity, hypokalemia, anemia	No
Liposomal amphotericin B	NA	Invasive aspergillosis, histoplasmosis: 3–5 mg/kg/day Cryptococcosis (meningoencephalitis, severe pneumonia): 3–4 mg/kg/day, or 10 mg/kg single dose Mucormycosis (off-label): 5–10 mg/kg/day	None	None	Infusion related toxicity (fever, chills, hypoxemia), nephrotoxicity, hypokalemia, anemia	No

TDM: requirement for therapeutic drug monitoring; CrCl: creatinine clearance; DR: delayed release; NA: not applicable.

## Data Availability

Not applicable.

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
