# Peer review of "Systemic Antifungal Therapy for Invasive Pulmonary Infections"

_jof, 2023, doi:10.3390/jof9020144_

Round 1
Reviewer 1 Report
It is superb, well written, excellent references (150 references). I recommend acceptance as is.
Author Response
Reviewer 1
It is superb, well written, excellent references (150 references). I recommend acceptance as is.
Response:
I thank the reviewer for their kind comments.
Reviewer 2 Report
J of F 2105072
SYSTEMIC ANTIFUNGAL THERAPY
This is an excellent review written by leading authority on the subject
I have some rather minor points for the authors to consider for their revision
The MS starts citing the references correctly in L 75, starting with ref 1. First 20 references lines 18-74 are cited incorrectly
Itraconazole. Consider making the point that there has been data suggest that azole antifungals 09in this situation itraconazole), through differential inhibition of hepatic cytochrome P-450 isoenzymes, affect metabolism of drugs used in condition regimen (with associated toxicities) and high vigilance is in order (see . Cyclophosphamide metabolism is affected by azole antifungals. Marr KA, Leisenring W, Crippa F, Slattery JT, Corey L, Boeckh M, McDonald GB.Marr KA, et al. Blood. 2004 Feb 15;103(4):1557-9. doi: 10.1182/blood-2003-07-2512. Epub 2003 Sep 22.Blood. 2004. PMID: 14504090 Free article. )
Consider emphasizing the efficacy the new mold-active triazoles (eg isavuconazole and voriconazole is suboptimal in neutropenic patients with opportunistic fungal infection (eg IA-see Impact of unresolved neutropenia in patients with neutropenia and invasive aspergillosis: a post hoc analysis of the SECURE trial. Kontoyiannis DP, Selleslag D, Mullane K, Cornely OA, Hope W, Lortholary O, Croos-Dabrera R, Lademacher C, Engelhardt M, Patterson TF. J Antimicrob Chemother. 2018 Mar 1;73(3):757-763.)
L 242: is there a reference to support the notion that hallucinations are associated with high Voriconazole (VRC) serum levels?
L 2546; Low GFR has been implicated in some ( but not all) patients with VRC associated fluorosis (Reversible skeletal disease and high fluoride serum levels in hematologic patients receiving voriconazole. Gerber B, Guggenberger R, Fasler D, Nair G, Manz MG, Stussi G, Schanz U. Blood. 2012 Sep 20;120(12):2390-4. )
L 264. consider emphasizing the fact that clinical trials excluded patients with breakthrough IPA on mold active prophylaxis (they enrolled patients on fluconazole or no prophylaxis0 and we do not have high grade evidence based on RCT on how to manage a common scenario, that of breakthrough IA on mold active agents (see discussion regarding the complexities of management of this entity in Breakthrough Invasive Mold Infections in the Hematology Patient: Current Concepts and Future Directions. Lionakis MS, Lewis RE, Kontoyiannis DPClin Infect Dis. 2018 Oct 30;67(10):1621-1630.
You might want to emphasize that azoles have Mucorales species specific differences in MIC 90s to posaconazole (POSA)
L 320; Higher doses of POSA have, surprisingly, significant interpatient variability in serum POSA levels ( see Effect of High-Dose Posaconazole on Serum Levels in Adult Patients with Hematologic Malignancy. DiPippo AJ, McDaneld PM, Tverdek FP, Kontoyiannis DP.DiPippo AJ, et al. Antimicrob Agents Chemother. 2021 Nov 17;65(12):e0123021. )
L 390 POSA has , as all drugs, very poor activity in persistently neutropenic patients with fusariosis (see Posaconazole as salvage treatment for invasive fusariosis in patients with underlying hematologic malignancy and other conditions. Raad II, Hachem RY, Herbrecht R, Graybill JR, Hare R, Corcoran G, Kontoyiannis DP.. Clin Infect Dis. 2006 May 15;42(10):1398-403. ).
L 407: Although MIICs do not correlate with outcome in Fusariosis (Do high MICs predict the outcome in invasive fusariosis? Nucci M, Jenks J, Thompson GR, Hoenigl M, Dos Santos MC, Forghieri F, Rico JC, Bonuomo V, López-Soria L, Lass-Flörl C, Candoni A, Garcia-Vidal C, Cattaneo C, Buil J, Rabagliati R, Roiz MP, Gudiol C, Fracchiolla N, Campos-Herrero MI, Delia M, Farina F, Fortun J, Nadali G, Sastre E, Colombo AL, Pérez Nadales E, Alastruey-Izquierdo A, Pagano L.Nucci M, et al. J Antimicrob Chemother. 2021 Mar 12;76(4):1063-1069. doi: 10.1093/jac/dkaa516.J Antimicrob Chemother. 2021. PMID: 33326585 ), isavuconazole has suboptimal in vitro efficacy in Fusarium species (In Vitro Susceptibility of Fusarium to Isavuconazole. Broutin A, Bigot J, Senghor Y, Moreno-Sabater A, Guitard J, Hennequin C.Broutin A, et al. Antimicrob Agents Chemother. 2020 Jan 27;64(2):). The real life data on isavuconazole efficacy in non-Aspergillus molds is promising, however data are hard to evaluate (Systemic antifungal therapy with isavuconazonium sulfate or other agents in adults with invasive mucormycosis or invasive aspergillosis (non-fumigatus): A multicentre, non-interventional registry study. Thompson GR 3rd, Garcia-Diaz J, Miceli MH, Nguyen MH, Ostrosky-Zeichner L, Young JH, Fisher CE, Clark NM, Greenberg RN, Spec A, Kovanda L, Croos-Dabrera R, Kontoyiannis DP.. Mycoses. 2022 Feb;65(2):186-198. doi: 10.1111/myc.13412. Epub 2021 Dec 22.Mycoses. 2022. PMID: 34888961).
L 518.A nice recent paper evaluate the time dependency and the effect comedications to of renal injury in patients receiving lipoAMB (Antimicrob Agents Chemother . 2017 Aug 24;61(9):e02651-16. doi: 10.1128/AAC.02651-16. Print 2017 Sep. Retrospective Cohort Analysis of Liposomal Amphotericin B Nephrotoxicity in Patients with Hematological Malignancies Marta Stanzani 1 , Nicola Vianelli 1 , Michele Cavo 1 , Alessandro Maritati 1 , Marta Morotti 2 , Russell E Lewis 3
L 556. ABLC appears to be well tolerated in patients with severe infusion reactions due to lipoAMB ( The safety of amphotericin B lipid complex in patients with prior severe intolerance to liposomal amphotericin B. Farmakiotis D, Tverdek FP, Kontoyiannis DP. Clin Infect Dis. 2013 Mar;56(5):701-3.
L 588. I would add the lipoAMB might also ne the preferred option in patients with breakthrough IA to azole prophylaxis
Anidulafungin appears to be well tolerated in patients with hepatotoxicity due to caspofungin Switching to anidulafungin from caspofungin in cancer patients in the setting of liver dysfunction is associated with improvement of liver function tests. Jung DS, Tverdek FP, Jiang Y, Kontoyiannis DP.. J Antimicrob Chemother. 2015 Nov;70(11):3100-6)
L 698. Ref 146 is cited incorrectly
Recent reviews try to define the future therapeutic “space “of the new antifungals and discuss the difficulties in understanding their true potential. Investigational Antifungal Agents for Invasive Mycoses: A Clinical Perspective. Lamoth F, Lewis RE, Kontoyiannis DP. Clin Infect Dis. 2022 Aug 31;75(3):534-544)

Author Response
Reviewer 2
This is an excellent review written by leading authority on the subject.
I thank the reviewer for their kind comments.
I have some rather minor points for the authors to consider for their revision
The MS starts citing the references correctly in L 75, starting with ref 1. First 20 references lines 18-74 are cited incorrectly
I thank the reviewer for noticing the error in reference formatting. The numbering has been corrected.
Itraconazole. Consider making the point that there has been data suggest that azole antifungals 09in this situation itraconazole), through differential inhibition of hepatic cytochrome P-450 isoenzymes, affect metabolism of drugs used in condition regimen (with associated toxicities) and high vigilance is in order (see . Cyclophosphamide metabolism is affected by azole antifungals. Marr KA, Leisenring W, Crippa F, Slattery JT, Corey L, Boeckh M, McDonald GB.Marr KA, et al. Blood. 2004 Feb 15;103(4):1557-9. doi: 10.1182/blood-2003-07-2512. Epub 2003 Sep 22.Blood. 2004. PMID: 14504090 Free article. )
A comment regarding increased toxicity during co-administration with cyclophosphamide was added, with the suggested reference (lines 155-158).
Consider emphasizing the efficacy the new mold-active triazoles (eg isavuconazole and voriconazole is suboptimal in neutropenic patients with opportunistic fungal infection (eg IA-see Impact of unresolved neutropenia in patients with neutropenia and invasive aspergillosis: a post hoc analysis of the SECURE trial. Kontoyiannis DP, Selleslag D, Mullane K, Cornely OA, Hope W, Lortholary O, Croos-Dabrera R, Lademacher C, Engelhardt M, Patterson TF. J Antimicrob Chemother. 2018 Mar 1;73(3):757-763.)
The effect of unresolved neutropenia on treatment success in the SECURE study was added (lines, 499-500).
L 242: is there a reference to support the notion that hallucinations are associated with high Voriconazole (VRC) serum levels?
A reference for elevated blood voriconazole levels in patients with hallucinations was added (New Ref 60: Zonios et al. Clin Infect Dis. 2008 July 1; 47(1): e7–e10).
L 2546; Low GFR has been implicated in some ( but not all) patients with VRC associated fluorosis (Reversible skeletal disease and high fluoride serum levels in hematologic patients receiving voriconazole. Gerber B, Guggenberger R, Fasler D, Nair G, Manz MG, Stussi G, Schanz U. Blood. 2012 Sep 20;120(12):2390-4. )
The possible role of impaired renal function was added to the section on voriconazole-induce fluorosis (lines 262-263).
L 264. consider emphasizing the fact that clinical trials excluded patients with breakthrough IPA on mold active prophylaxis (they enrolled patients on fluconazole or no prophylaxis0 and we do not have high grade evidence based on RCT on how to manage a common scenario, that of breakthrough IA on mold active agents (see discussion regarding the complexities of management of this entity in Breakthrough Invasive Mold Infections in the Hematology Patient: Current Concepts and Future Directions. Lionakis MS, Lewis RE, Kontoyiannis DPClin Infect Dis. 2018 Oct 30;67(10):1621-1630.
The point regarding the limited data to support treatment options for breakthrough IFD is highlighted in lines 43-45, where the paper by Lionakis and Kontoyiannis is also cited (Ref. 15).
You might want to emphasize that azoles have Mucorales species specific differences in MIC 90s to posaconazole (POSA)
L 320; Higher doses of POSA have, surprisingly, significant interpatient variability in serum POSA levels ( see Effect of High-Dose Posaconazole on Serum Levels in Adult Patients with Hematologic Malignancy. DiPippo AJ, McDaneld PM, Tverdek FP, Kontoyiannis DP.DiPippo AJ, et al. Antimicrob Agents Chemother. 2021 Nov 17;65(12):e0123021. )
A note regarding interpatient variability among patients receiving high dose oral DR posaconazole was added (lines 341-342).
L 390 POSA has , as all drugs, very poor activity in persistently neutropenic patients with fusariosis (see Posaconazole as salvage treatment for invasive fusariosis in patients with underlying hematologic malignancy and other conditions. Raad II, Hachem RY, Herbrecht R, Graybill JR, Hare R, Corcoran G, Kontoyiannis DP.. Clin Infect Dis. 2006 May 15;42(10):1398-403. ).
The effect of unresolved neutropenia on treatment success was added to the section on fusariosis (lines 412-415).
L 407: Although MIICs do not correlate with outcome in Fusariosis (Do high MICs predict the outcome in invasive fusariosis? Nucci M, Jenks J, Thompson GR, Hoenigl M, Dos Santos MC, Forghieri F, Rico JC, Bonuomo V, López-Soria L, Lass-Flörl C, Candoni A, Garcia-Vidal C, Cattaneo C, Buil J, Rabagliati R, Roiz MP, Gudiol C, Fracchiolla N, Campos-Herrero MI, Delia M, Farina F, Fortun J, Nadali G, Sastre E, Colombo AL, Pérez Nadales E, Alastruey-Izquierdo A, Pagano L.Nucci M, et al. J Antimicrob Chemother. 2021 Mar 12;76(4):1063-1069. doi: 10.1093/jac/dkaa516.J Antimicrob Chemother. 2021. PMID: 33326585 ), isavuconazole has suboptimal in vitro efficacy in Fusarium species (In Vitro Susceptibility of Fusarium to Isavuconazole. Broutin A, Bigot J, Senghor Y, Moreno-Sabater A, Guitard J, Hennequin C.Broutin A, et al. Antimicrob Agents Chemother. 2020 Jan 27;64(2):). The real life data on isavuconazole efficacy in non-Aspergillus molds is promising, however data are hard to evaluate (Systemic antifungal therapy with isavuconazonium sulfate or other agents in adults with invasive mucormycosis or invasive aspergillosis (non-fumigatus): A multicentre, non-interventional registry study. Thompson GR 3rd, Garcia-Diaz J, Miceli MH, Nguyen MH, Ostrosky-Zeichner L, Young JH, Fisher CE, Clark NM, Greenberg RN, Spec A, Kovanda L, Croos-Dabrera R, Kontoyiannis DP.. Mycoses. 2022 Feb;65(2):186-198. doi: 10.1111/myc.13412. Epub 2021 Dec 22.Mycoses. 2022. PMID: 34888961).
The suggested reference on in vitro susceptibility of Fusarium species was added to the section on isavuconazole (lines 444-446, New Ref 106).
L 518.A nice recent paper evaluate the time dependency and the effect comedications to of renal injury in patients receiving lipoAMB (Antimicrob Agents Chemother . 2017 Aug 24;61(9):e02651-16. doi: 10.1128/AAC.02651-16. Print 2017 Sep. Retrospective Cohort Analysis of Liposomal Amphotericin B Nephrotoxicity in Patients with Hematological Malignancies Marta Stanzani 1 , Nicola Vianelli 1 , Michele Cavo 1 , Alessandro Maritati 1 , Marta Morotti 2 , Russell E Lewis 3
The reference on the observed combined nephrotoxicity of amphotericin B and cyclosporine was added in line 561 (New Ref 124).
L 556. ABLC appears to be well tolerated in patients with severe infusion reactions due to lipoAMB ( The safety of amphotericin B lipid complex in patients with prior severe intolerance to liposomal amphotericin B. Farmakiotis D, Tverdek FP, Kontoyiannis DP. Clin Infect Dis. 2013 Mar;56(5):701-3.
The reference on the tolerability of ABLC in patients with previous LAMB infusion reaction was added in lines 595-596 (New Ref 131).
L 588. I would add the lipoAMB might also ne the preferred option in patients with breakthrough IA to azole prophylaxis
The role of LAMB as treatment of breakthrough IFD is discussed in lines 45-47.
Anidulafungin appears to be well tolerated in patients with hepatotoxicity due to caspofungin Switching to anidulafungin from caspofungin in cancer patients in the setting of liver dysfunction is associated with improvement of liver function tests. Jung DS, Tverdek FP, Jiang Y, Kontoyiannis DP.. J Antimicrob Chemother. 2015 Nov;70(11):3100-6)
The safety of anidulafungin in patients who experienced hepatotoxicity with caspofungin treatment was added in lines 714-716.
L 698. Ref 146 is cited incorrectly
I thank the reviewer for noticing this error. The reference was moved to the section on echinocandins and invasive aspergillosis. Both sections on IPA and CPA were restructured to accommodate this change.
Recent reviews try to define the future therapeutic “space “of the new antifungals and discuss the difficulties in understanding their true potential. Investigational Antifungal Agents for Invasive Mycoses: A Clinical Perspective. Lamoth F, Lewis RE, Kontoyiannis DP. Clin Infect Dis. 2022 Aug 31;75(3):534-544)
The thoughtful review by Lamoth et al. was added as a reference to the closing section (New Ref. 172).